# Effect of Aminosilane Coupling Agent-Modified Nano-SiO₂ Particles on Thermodynamic Properties of Epoxy Resin Composites

**Gang Lv [1], Ke Li [1], Yubing Shi [2], Ruiliang Zhang [1], Huadong Tang [1] and Chao Tang [3,*]**

[1]  Guiyang Bureau of EHV Power Transmission Company, China Southern Power Grid Co. Ltd.,
     Guiyang 550000, China; lvgang@ehv.csg.cn (G.L.); like@ehv.csg.cn (K.L.); Zhangruiliang@ehv.csg.cn (R.Z.);
     tanghuadong@ehv.csg.cn (H.T.)
[2]  Nanjing Electric High Voltage Bushing Co. Ltd., Nanjing 210038, China; shiyubing@ld-cn.com
[3]  College of Engineering and Technology, Southwest University, Chongqing 400715, China
[*]  Correspondence: swutc@swu.edu.cn

**Abstract:** From the perspective of improving the thermodynamic properties of epoxy resin, it has become the focus of research to enhance the operational stability of GIS (Gas Insulated Substation) basin insulators for UHV (Ultra-High Voltage) equipment. In this paper, three aminosilane coupling agents with different chain lengths, (3-Aminopropyl)trimethoxysilane (KH550), Aminoethyl)-γ-aminopropyltrimethoxysilane (KH792) and 3-[2-(2-Aminoethylamino)ethylamino]propyl-trimethoxysilane (TAPS), were used to modify nano-SiO₂ and doped into epoxy resin, respectively, using a combination of experimental and molecular dynamics simulations. The experimental results showed that the surface-grafted KH792 model of nano-SiO₂ exhibited the most significant improvement in thermal properties compared with the undoped nanoparticle model. The storage modulus increased by 276 MPa and the Tg increased by 61 K. The simulation results also showed that the mechanical properties of the nano-SiO₂ surface-grafted KH792 model were about 3 times higher than that of the undoped nanoparticle model, the Tg increased by 36.5 K, and the thermal conductivity increased by 24.5%.

**Keywords:** UHV equipment; GIS; basin insulator; epoxy resin; silane coupling agent; thermal properties

## 1. Introduction

In recent years, the national UHV power grid project has developed rapidly. The GIS is an important piece of equipment for ensuring the safe and stable operation of a UHV power grid [1]. As key equipment in the GIS, the properties of basin insulators have an important impact on the safe and stable operation of high-voltage insulation equipment [2]. Therefore, it is crucial to design a composite material that meets the requirements. Epoxy resin, as the main raw material of basin insulators, is easy to be processed. It is low cost and has good corrosion resistance and electrical insulation properties [3]. In recent years, with the booming development of high-voltage technology, the voltage level is getting higher and higher, so the requirements for the properties of voltage equipment are also getting higher and higher. Therefore, improving the thermal properties of epoxy resin is of great significance for the stable operation and practical application of basin insulators.

It was found that doping nanoparticles can effectively improve the thermal properties of polymer materials [4–6]. Nanoparticles themselves have a relatively large specific surface area [7], so when doping nanoparticles directly, the phenomenon of agglomeration occurs easily, which makes nanoparticles fail to disperse well into the matrix, and thus, the expected effect cannot be achieved. Domestic and foreign scholars find that grafting silane coupling agents on the surface of nanoparticles can effectively improve the phenomenon of agglomeration and the properties of polymers. Hang Zhao [8] et al. found that the conversion sensitivity of composites obviously increases by 57.4% after the modification of

nanocomposites with silane coupling agents. In the literature [9], the silane coupling agent KH560 has been used to modify nano aluminum oxide, which effectively improved the mechanical properties and electrical strength of epoxy resin. Na Wang discovered through experiments that the tensile strength and elastic modulus were increased by 99.2% and 110%, respectively, after a silane coupling agent grafted the nanoparticles. Studies [10–13] have shown that the surface treatment of nanoparticles by coupling agents can effectively improve the thermal properties of them.

With the continuous development of computer technology, the molecular dynamics method, as a new research tool, has received extensive attention from scholars at home and abroad in recent years [14–18]. Molecular Simulation (MS) techniques can not only validate macroscopic experiments from a microscopic perspective but also predict and probe the properties of materials, reducing the trial-and-error efforts and time costs of traditional experiments [19]. In recent years, domestic and foreign experts have made great developments and advances in different fields and disciplines using molecular simulation techniques [20–23]. S.C. Chowdhury et al. used molecular dynamics simulations to study the interaction of epoxy resins with glass fiber in the presence of monolayer glycidoxypropyl-trimethoxy silane. The results showed that the presence of silane introduces covalent bonding interactions in the fiber–epoxy interphase, improving the interphase properties [24]. W.Q. Zhang et al. used molecular dynamics simulations to explore the ideal structure design of graphene oxide (GO) modified by poly (dopamine) (PDA), which is more conducive to the performance improvement of epoxy resin composites [25].

The silane coupling agent selected in this article is a commonly used silane coupling agent and a typical coupling agent [26–28]. As an amphiphilic modifier, the silane coupling agent can react with the hydroxyl group on the surface of the nanoparticle to form a strong chemical bond; the other end can be physically entangled or chemically reacted with the matrix polymer, thus firmly bonding two less compatible materials together and building a "molecular bridge" between the nanoparticle and the polymer with special functions. It can also inhibit the separation of the nanoparticle and polymer composite system "phases", increasing the filling amount and maintaining good dispersion, which can significantly improve the mechanical properties, and impact the strength and flexibility of the polymer [29,30].

We used the experimental means and molecular dynamics methods to study the effects of the nanoparticles modified with three aminosilane coupling agents of different chain lengths on the thermal properties of epoxy resin composites, and the better performing aminosilane coupling agents were selected by calculating the characteristic parameters, such as the storage modulus, mechanical properties, glass transition temperature, and thermal conductivity. The experimental results are of great reference value for the improvement of the thermal properties of basin insulators.

## 2. Experimental and Simulation Methods

### 2.1. Materials Preparation

Matrix materials: bisphenol A epoxy resin (E44), curing agent: m-phenylenediamine (mPD). The molecular structure of epoxy resin is shown in Figure 1. Inorganic nanoparticles: nano silica particles with a particle size of 35 nm ($SiO_2$), dispersing agent: ethanol (with a density of 1.03–1.058 g/cm$^3$ at room temperature), silane coupling agent: KH550, KH792, TAPS. Figure 2 shows the diagrammatic sketch of the three silane coupling agents grafted onto the surface of nano silica particles.

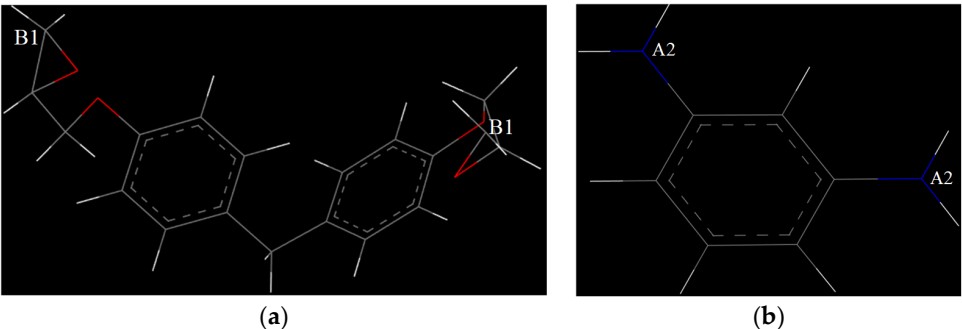

**Figure 1.** (**a**) Bisphenol A epoxy resin; (**b**) m-phenylenediamine curing agent.

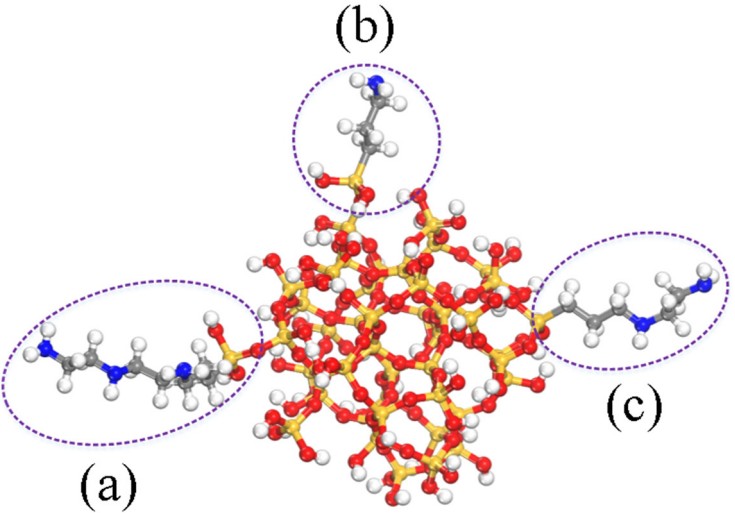

**Figure 2. The** three kinds of silane coupling agents grafted on the surface of nano-silica particles, (**a**) TAPS, (**b**) KH550, (**c**) KH792.

*2.2. Experimental Preparation*

2.2.1. Surface Grafting Treatment of Nano Silica Particles

A certain amount of anhydrous ethanol was added to a stirrer, and then the silane coupling agent KH550 was slowly added by dripping. Then, the $SiO_2$ nanoparticles were slowly added, stirred well, and dispersed with ultrasonic waves. Finally, they were washed and dried with anhydrous ethanol to obtain the KH550-treated nanoparticles. We obtained the KH792-treated and the TAPS-treated nanoparticles in the same way.

2.2.2. Preparation of Epoxy Resin Composites

We labeled the pure epoxy resin, the epoxy resin doped with pure nano silica particles, and the samples prepared by doping the epoxy resin with the nanoparticles treated with KH550, KH792, and TAPS as a, b, c, d, and e, respectively, as shown in Table 1. Then we weighed an appropriate amount of curing agents, added them to the samples, and mixed well. After curing, strip samples were prepared. The curing conditions were 100 °C/2 h, 110 °C/2 h, and 120 °C/2 h.

**Table 1.** Test samples.

| Number | Samples |
|---|---|
| a | DGEBA |
| b | DGEBA + $SiO_2$ |
| c | DGEBA + $SiO_2$ + KH550 |
| d | DGEBA + $SiO_2$ + KH792 |
| e | DGEBA + $SiO_2$ + TAPS |

### 2.3. Simulation Methods

We used materials simulation software for model building and parameter calculation. Pure epoxy resin model, a nano-silicone epoxy resin composite model, and a nano-silicone epoxy resin composite model with surface grafting of KH550, KH792, and TAPS were built using the Amorphous tool. The Forcite module was used to optimize and anneal the constructed model. (1) In the Amorphous cell module, we added DGEBA and MPD by a ratio of 2:1 and set the density as 0.6 g/$cm^3$. We chose the COMPASS force field as the calculating force field [31,32] and built the three-dimensional periodic box. After geometrically optimizing the constructed model, we first performed the 500 ps NVT calculation (the particle number N, volume V, and temperature T of the system remain constant), and then performed the 500 ps NPT calculation (the particle number N, pressure P, and temperature T of the system remain constant). Finally, we used the Perl script for crosslinking, presetting the crosslinking degree as 95%. (2) We chose the nano-$SiO_2$ particle with a particle size of 10 nm for hydrogenation, built a nano-box of $50 \times 50 \times 50$ Å$^3$, and built the model according to the method in (1). (3) We grafted KH550, KH792, and TAPS on the surfaces of the nanoparticles hydrogenated in (2), respectively. The grafting density on the surface of nanoparticles was 10%. The model was built according to the steps in (1). A Nosé thermostat was used for temperature control, and the Berendsen method was used for pressure control.

### 2.4. Tests

We used the dynamic thermal mechanical analyzer (DMA) to analyze the storage modulus and glass transition temperature of the five different groups of samples. The test samples were $60 \times 8 \times 5$ mm$^3$ rectangles. The load applied in the test was 10 N, while the heating rate was kept at 2 °C/min. The frequency applied in the test was 1 Hz, and the temperature range of the five groups of sample tests was mainly 0–550 K.

## 3. Results and Discussion

### 3.1. Experimental Results

#### 3.1.1. Storage Modulus

Figure 3 shows the storage modulus of the five models at different temperatures. It can be seen that the storage modulus of the five groups of models decreased as the temperature increased. In the beginning, the decrease was relatively gentle. When it reached a certain temperature, a sharp decrease appeared, which should be related to the glass transition temperature of materials. The storage modulus of the five models a, b, c, d, and e at 300 K are 1680 MPa, 1720 MPa, 1879 MPa, 1956 MPa, and 1915 MPa, respectively. After the silane coupling agent was grafted onto the surface of the nanoparticles, the storage modulus was effectively increased, among which the grafted KH792 had the most obvious improvement of 276 MPa over the pure epoxy resin.

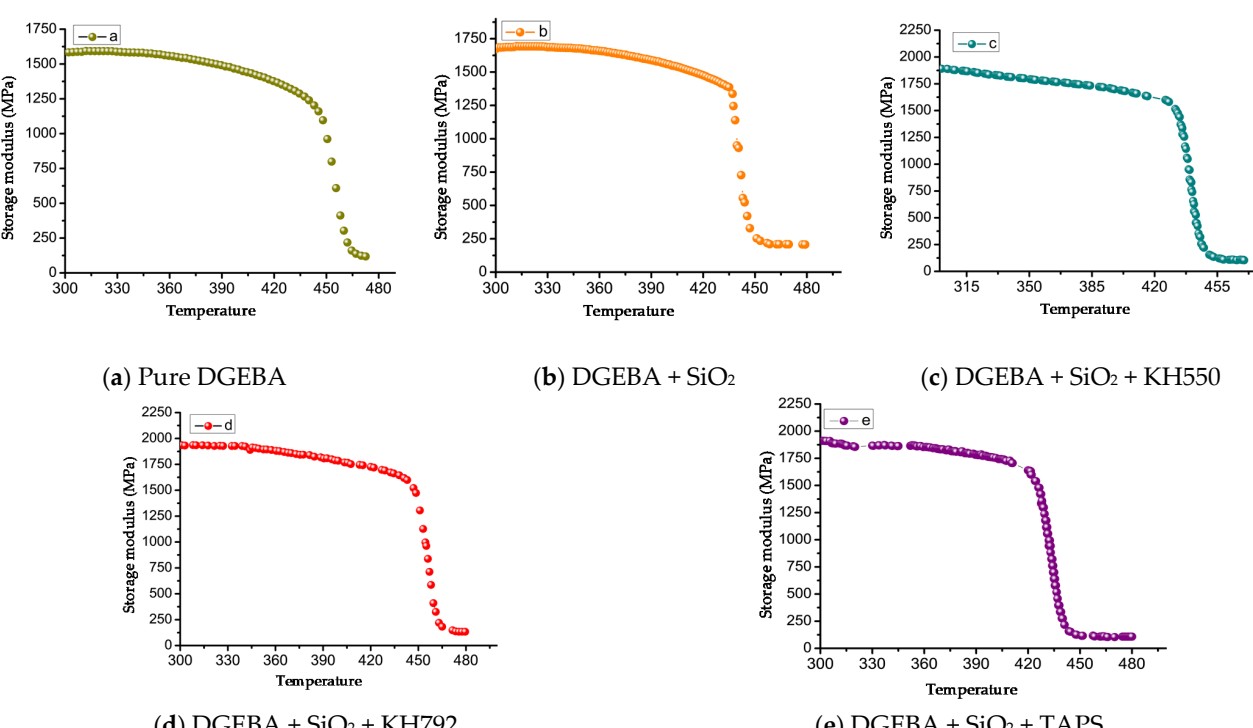

**(a)** Pure DGEBA     **(b)** DGEBA + SiO$_2$     **(c)** DGEBA + SiO$_2$ + KH550

**(d)** DGEBA + SiO$_2$ + KH792     **(e)** DGEBA + SiO$_2$ + TAPS

**Figure 3.** Storage modulus of five samples at the same temperature.

It has been shown that the number of hydrogen bonds formed between SiO$_2$ nanoparticles modified on the surface of silane coupling agents and epoxy resins is greater than the number of hydrogen bonds formed between unmodified SiO$_2$ nanoparticles and epoxy resins [33]. The formation of hydrogen bonds enhances the compatibility between SiO$_2$ nanoparticles and epoxy resin and improves the dispersion stability of the nanoparticles in the epoxy resin. Therefore, the surface-modified nano-SiO$_2$/epoxy composites have a large energy storage modulus.

### 3.1.2. Mechanical Loss Factor

Figure 4 shows the variation curves of the mechanical loss factor of the five models tested under different temperatures. The peak value in the figure is the glass transition temperature of the corresponding model.

### 3.2. Simulation Results

### 3.2.1. Calculation of Mechanical Properties

The model mechanical property parameters of the five groups of models were measured using the static principle method. The elastic modulus E, bulk modulus K, and shear modulus G of the composite model can be calculated by the following formula:

$$E = \frac{\mu(3\lambda + 2\mu)}{\lambda + \mu} \tag{1}$$

$$K = \lambda + \frac{2}{3}\mu \tag{2}$$

$$G = \mu \tag{3}$$

where $\lambda$ is the Lame constant. Table 2 shows the mechanical property parameters of the five models at 300 K. According to the statistical results, the mechanical properties in three models, c, d, and e, grafted with the silane coupling agents are higher than those in the pure model a and the directly doped nano-silica particle model b. Among them, the grafting of KH792 on the surface of the nanoparticles had the most obvious effect of increasing the

mechanical properties by about 3 times compared with the pure epoxy resin. The research results show that the grafting of silane coupling agent on the surface of nanoparticles can effectively increase the mechanical properties of epoxy resin, which is probably because the coupling agent improves the dispersion of nanoparticles and makes the nanoparticles fully contact with the epoxy resin, thus improving the mechanical properties of the epoxy resin.

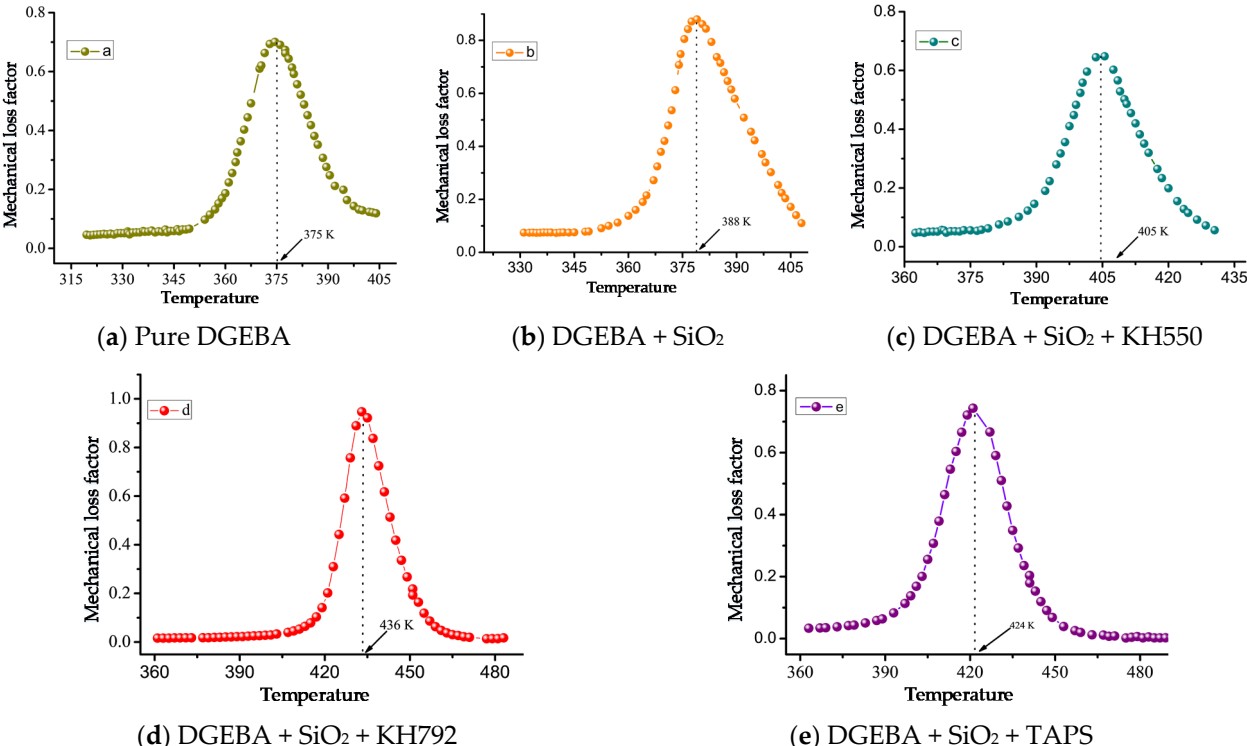

**Figure 4.** Mechanical loss factor for five samples under different temperatures.

**Table 2.** Mechanical properties parameters (GPa).

|   | a | b | c | d | e |
|---|---|---|---|---|---|
| E | 4.92 | 5.53 | 6.21 | 7.28 | 6.53 |
| K | 4.66 | 4.72 | 5.69 | 6.86 | 6.11 |
| G | 1.82 | 2.17 | 2.46 | 2.85 | 2.54 |

### 3.2.2. Calculation of Glass Transition Temperature

The glass transition temperatures of the polymers can be calculated from the interaction energy [34,35]. Figure 5 shows the glass transition temperatures of the $SiO_2$, A1100, A1120, and A1130 models calculated through the linear fitting relationship between non-bond energies and temperatures. The simulation results are shown in Table 3.

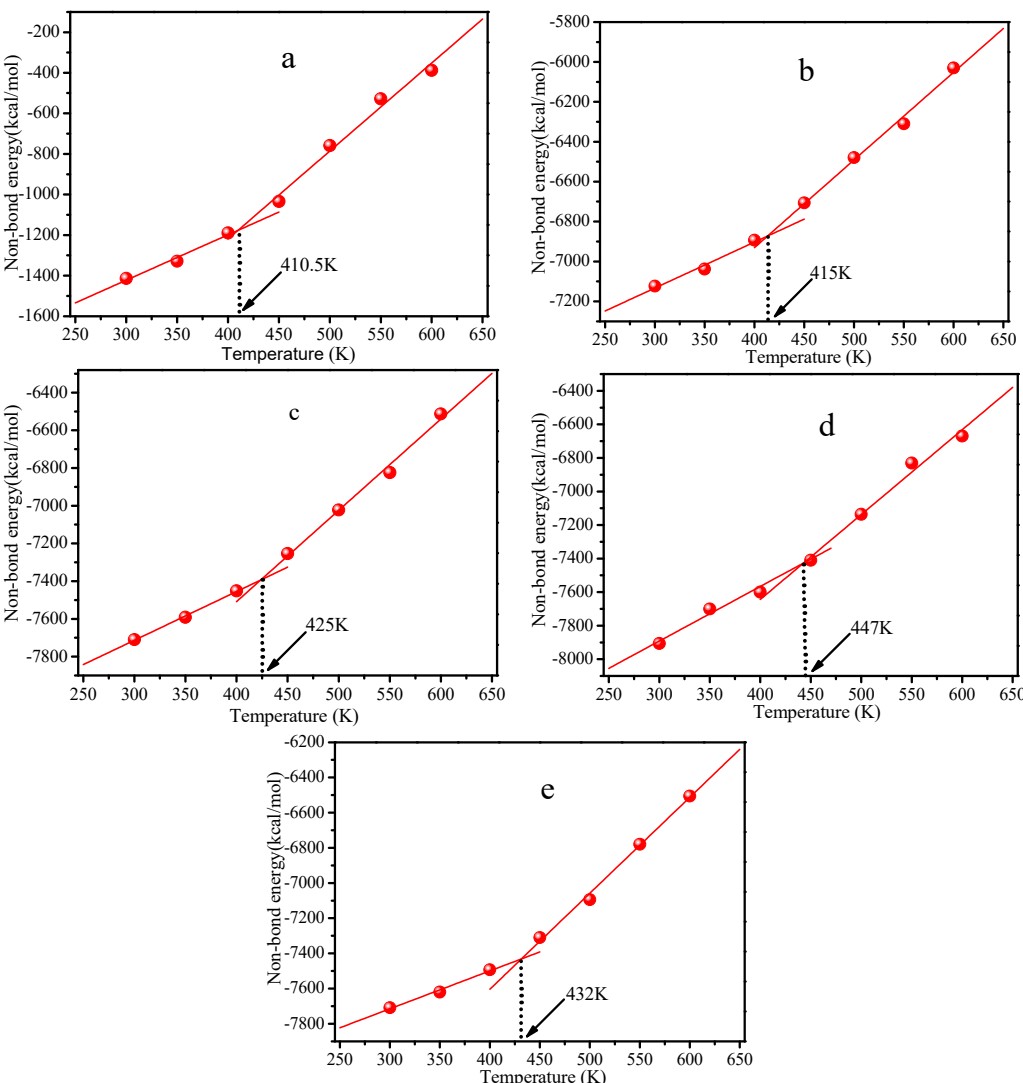

**Figure 5.** Glass transition temperatures: (**a**) Pure DGEBA, (**b**) DGEBA + SiO$_2$, (**c**) DGEBA + SiO$_2$ + KH550, (**d**) DGEBA + SiO$_2$ + KH792, (**e**) DGEBA + SiO$_2$ + TAPS.

**Table 3.** Glass transition temperature (K).

|  | a | b | c | d | e |
|---|---|---|---|---|---|
| Experimental Values | 375 | 388 | 405 | 436 | 428 |
| Simulation Values | 410.5 | 415 | 425 | 447 | 432 |

The glass transition temperatures calculated in Table 3 range from 388 K to 447 K. The results are consistent with those calculated in the references [17,21,36,37], which shows that the calculation results in this paper are correct. It was found that the experimental calculation values are lower than the simulated values, which may have been caused by the experimental environment, but the variation trend is consistent. Among the five models, silane coupling agent grafting on the nanoparticle surface resulted in a significant increase in the glass transition temperature, while grafting KH792 on the nanoparticle surface had the most significant increase. Compared with the pure epoxy resin, the temperature increased by 44 K and 36.5 K, respectively, in the experimental and simulation results.

### 3.2.3. Thermal Conductivity

Thermal conductivity can be used to characterize the thermal properties of materials. We used the non-equilibrium dynamics method to calculate the thermal conductivity [38,39]. The calculation formula is as follows:

$$\lambda = \frac{1}{3Vk_BT^2} \int_0^\infty \langle J(t)J(0)\rangle dt \tag{4}$$

where $V$ is the volume, $k_B$ is the Boltzman constant, $T$ is the temperature, $t$ is the time, and $J$ is the micro heat flow in the system. Then, a perturbation was applied to the system to establish the nonequilibrium thermal conductivity process. Finally, the thermal conductivity was calculated using Fourier's law of heat conduction, as shown below:

$$J = -\lambda VT \tag{5}$$

where $VT$ is the constant temperature difference. Table 4 shows the thermal conductivity of the five models at 300 K. It can be found that the thermal conductivity of crosslinked epoxy resin increased to some degree after adding the nanoparticles. Among them, the thermal conductivity of the $SiO_2$ model increased by 0.9%. This increasing range is relatively small. However, the increase in thermal conductivity of the nanocomposites modified by the silane coupling agents was relatively high, and the thermal conductivities of KH550, KH792, and TAPS increased by 4.9%, 24.5%, and 3.2%, respectively. These results further show that the silane coupling agent-modified nanoparticles can improve the thermal properties of cross-linked epoxy resin.

**Table 4.** Calculations of the thermal conductivity of different models at 300 K (W·(m·K) $^{-1}$).

|  | a | b | c | d | e |
|---|---|---|---|---|---|
| Thermal Conductivity | 0.212 | 0.214 | 0.223 | 0.264 | 0.219 |

As can be seen from Figure 2, the main differences between these three silane coupling agents are the following: KH550 contains only one amino group ($-NH_2$), KH792 contains one imino group ($-NH-$) and one amino group ($-NH_2$), and TAPS contains two imino groups ($-NH-$) and one amino group ($-NH_2$). These three coupling agents contain imino, amino, and other groups, and the N atoms in imino and amino can form hydrogen bonds with epoxy resin molecules; therefore, the presence of the silane coupling agent improves the dispersion stability of nano-$SiO_2$ in epoxy resin, which is beneficial to improving the storage modulus, the glass transition temperature, the thermal conductivity, and other properties of nano-$SiO_2$/epoxy composite materials.

KH792 contains both imino and amino groups, while KH550 only contains amino groups; therefore, the number of hydrogen bonds formed between nano-$SiO_2$ modified by KH792 and epoxy resin is greater than that formed between nano-$SiO_2$ modified by KH550 and epoxy resin. Moreover, the comprehensive performance of nano-$SiO_2$/epoxy resin composite modified by KH792 is better than that of nano-$SiO_2$/epoxy composite modified by KH550. In addition, although TAPS contains two imino groups and one amino group, the chain length of TAPS is longer than that of KH792. Therefore, TAPS will form a thicker coating layer on the surface of nano-$SiO_2$, forming a relatively large steric hindrance, which will reduce the intermolecular bonding between nano-$SiO_2$ and epoxy resin and is not conducive to improving the thermodynamic properties of epoxy resin.

### 4. Conclusions

This study combined experimental and molecular dynamics simulations to study the thermal properties of epoxy resin of silica nanoparticles modified with three silane coupling agents, KH550, KH792, and TAPS. The conclusions are as follows:

The doped silane coupling agent-modified silica nanoparticles can effectively improve the thermal properties of epoxy resin, as well as the storage modulus, mechanical properties, glass transition temperature, and thermal conductivity of epoxy resin. Among the three silane coupling agents used in this study, KH792 had the most obvious synergistic effect. The experimental results show that the storage modulus of KH792 increased by 276 MPa compared with the undoped pure epoxy resin. Meanwhile, the results of the molecular dynamics simulation also show that the three groups of mechanical properties of KH792 increased by about 3 times, and the thermal conductivity increased by 24.5% compared with that of undoped pure epoxy resin.

The research results of this paper show that the modification of epoxy resin by silane coupling agent-modified nanoparticles can significantly improve their thermal properties, and choosing the silane coupling agent with a suitable chain length is particularly important for the final modification effect. The improvement of thermal properties of epoxy resin can enhance the operational stability of basin insulators under UHV conditions, and better ensure the safe and stable operation of a UHV project.

**Author Contributions:** Conceptualization, G.L. and C.T.; methodology, K.L. and H.T.; software, R.Z.; validation, C.T., writing—original draft preparation, G.L. and C.T.; writing—review and editing, G.L., Y.S. and C.T. All authors have read and agreed to the published version of the manuscript.

**Funding:** This research was funded by the science and technology project of EHV Power Transmission Company, China Southern Power Grid.

**Data Availability Statement:** The data that support the findings of this study are available from the corresponding author, C.T., upon reasonable request.

**Conflicts of Interest:** The authors declare no conflict of interest.

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
