# Peer review of "Effect of Aminosilane Coupling Agent-Modified Nano-SiO2 Particles on Thermodynamic Properties of Epoxy Resin Composites"

_processes, doi:10.3390/pr9050771_

Round 1

Reviewer 1 Report

The paper describes the effect of coupling agents on thermal properties of epoxy resin for GIS application.

Some points need clarification:

It would have been of interest, why these coupling agents were chosen. What is a typical coupling agent and where are main differences? Is there an explaination, why KH792 shows the best results? Here, a discussion of the different coupling agents should be added.

The language of the paper should be improved, especially in chapter 2 and within the first paragraph of chapter 3. Furthermore, the labels for figures and tables are often too general. Please give more details about the shown content. The labelling of the axes should be in English. Please check the layout of the text as well.

Author Response

Thanks for your comments.

Reviewer 2 Report

Comments:

Abstract –

  1. be-come -> become
  2. Please use full form before using abbreviations such as GIS, UHV, KH, TAPS
  3. English is not satisfactory in some parts

Introduction –

  1. The introduction section is very short and inadequate to give a detailed background.
  2. Include some of these works on nanofiller agglomerations. https://doi.org/10.1016/j.jcis.2018.08.105, https://doi.org/10.1002/pen.25045

Experiment –

  1. Why the results of DMA are shown in the experimental section, they should be in the Result section.
  2. The storage modulus of 5 different epoxies is stated but the reason, why the improvement in storage modulus happened, is not explained.

Simulation test –

  1. The details of the working principle of the material simulation software are missing. For the readers, who are not familiar with material modeling, it will be hard to follow the process.

Author Response

Thanks for your comments.

Round 2

Reviewer 2 Report

The authors answered my questions satisfactorily. The manuscript can be accepted for publication. 

Author Response

Thanks for your comments. At the time of revised manuscript submission, we have invited professional organizations to polish the full text.